# Ray Tracing Method of Gradient Refractive Index Medium Based on Refractive Index Step

Qingpeng Zhang [1,2,3], Yi Tan [1,2,3,*], Ge Ren [1,2,3] and Tao Tang [1,2,3]

1    Institute of Optics and Electronics, Chinese Academy of Sciences, No. 1 Guangdian Road, Chengdu 610209, China; zhangqingpeng16@mails.ucas.ac.cn (Q.Z.); renge@ioe.ac.cn (G.R.); tangtao24@163.com (T.T.)
2    Key Laboratory of Optical Engineering, Chinese Academy of Sciences, Chengdu 610209, China
3    University of the Chinese Academy of Sciences, Beijing 100049, China
*    Correspondence: tandeman@126.com; Tel.: +86-13153917751

**Featured Application: This method is mainly applied to gradient index media with a sudden change in refractive index or a large range of refractive index gradient changes, such as around the shock layer of an aircraft.**

**Abstract:** For gradient refractive index media with large refractive index gradients, traditional ray tracing methods based on refined elements or spatial geometric steps have problems such as low tracing accuracy and efficiency. The ray tracing method based on refractive index steps proposed in this paper can effectively solve this problem. This method uses the refractive index step to replace the spatial geometric step. The starting point and the end point of each ray tracing step are on the constant refractive-index surfaces. It avoids the problem that the traditional tracing method cannot adapt to the area of sudden change in the refractive index and the area where the refractive index changes sharply. Therefore, a suitable distance can be performed in the iterative process. It can achieve high-efficiency and precise ray tracing in areas whether the refractive index changes slowly or sharply. According to the comparison of calculation examples, this method can achieve a tracing accuracy of $10^{-5}$ mm. The speed and precision of ray tracing are better than traditional methods.

**Keywords:** gradient index media 1; ray tracing 2; refractive index step 3; constant-index surface 4; law of refraction 5

## 1. Introduction

The optical phenomenon in the gradient refractive index medium, as a ubiquitous objective physical phenomenon in nature, has been noticed for a long time. The Maxwell fisheye lens, the Wood lens, the Luneburg lens, the Gradient index rod lens, etc., can all be regarded as historical signs of the research on gradient index optics [1]. Due to the change in the refractive index of the medium, the original path of the light propagating in it is changed, which results in the deflection or phase change of the light and it also results in the image offset, blur, and jitter. Therefore, ray tracing in gradient refractive index media has become a research hotspot.

The ray tracing through graded refractive index media is based on the solution of the ray formula:

$$\frac{d}{ds}\left(n\frac{d\mathbf{r}}{ds}\right) = \nabla n \tag{1}$$

where $s$ is the distance along the ray path, $\mathbf{r}$ is the position of the ray, and $n$ is the index of refraction. Regarding Formula (1), Lucian proposed a universal numerical solution in 1968 [2]. Since then, the Runge–Kutta method, Euler method, and Taylor series expansion method have been applied to solve Formula (1), which have laid the foundation of gradient refractive index media ray tracing for a long time [3,4]. Most ray tracing methods are based on the three methods which have been listed on references [5–7]. In the ray tracing process,

these methods are all iteratively based on the refined elements [8–10] or spatial geometric steps [4,11,12]. When ray tracing is performed in a gradient refractive index medium with a gentle refractive index gradient, the accuracy of the ray tracing can be controlled by changing the geometric step. However, when encountering a gradient refractive index field with a sharp change (such as the shock wave of a high-speed aircraft [13–15]), it is necessary to adjust the geometric step according to the refractive index gradient aimed to be self-adaptive to the local refraction index gradient and grid geometry scale [7,16]. Even so, it is still possible to cross the region where the refractive index changes sharply in one iterative step. This situation often causes the refraction effect of the refractive index layer whose refractive index changes sharply to be ignored. In order to ensure the accuracy of the entire ray tracing path, the geometrical step-based ray tracing method (RTSGS) generally needs to set a smaller geometric tracing step. It will decrease efficiency in the tracing area where the refractive index changes slowly. Therefore, it will increase useless calculations [17].

This paper proposes a ray tracing method based on refractive index steps (RTRIS). In this method, the refractive index step $\Delta n$ is used to replace the geometric step $\Delta s$ of the traditional ray tracing method. The advantage of this method is that each iterative process takes equal refractive index step $\Delta n$ as the iterative step. In areas where the refractive index changes slowly, a relatively large distance will be performed in the iterative process to achieve high-efficiency ray tracing. In areas where the refractive index changes sharply or suddenly, the iterative step will reduce automatically to capture the sudden changes in the refractive index effectively. It can achieve refined tracing and improve ray tracing accuracy. The application of this method avoids the problem of being unable to capture regions of sudden changes in the refractive index after applying the step size regulation function. It can also effectively avoid the problem of insufficient tracing accuracy or low efficiency caused by unreasonable geometric step setting. By using the proposed method, the efficiency of ray tracing can be achieved while the accuracy of ray tracing can be improved. Thus, the method realizes the adaptive process of the gradient index medium in the ray tracing process.

## 2. The Method of Ray Tracing Based on Refractive Index Step

According to the differential geometry thought, the gradient index medium is divided into many small areas by the constant index surface. The medium between two constant-index surfaces is regarded as an equal refractive index medium. The value of the refractive index for this medium is replaced as the average of the two constant-index surfaces. Those constant-index surfaces are regarded as refractive surfaces. Snell's law is used to calculate the direction vector of the refracted ray after the ray passes through the constant index surface. So, Snell's law is the foundation of RTSGS [15].

### 2.1. Space Snell's Law

Snell's law in vector form is much more useful in the process of ray tracing in three-dimensional space [18]. According to the reference [6], we can calculate Snell's law in vector form.

The plane Snell's law can usually be expressed as:

$$n' \sin I' = n \sin I \tag{2}$$

In this formula, $n$ and $n'$ are the refractive index of the medium on the incident side and refracted side, respectively. $I$ and $I'$ are the incident angle and refraction angle of the ray, respectively.

Combined with Figure 1, Formula (2) can be modified as:

$$n'(\mathbf{A}_0' \times \mathbf{N}) = n(\mathbf{A}_0 \times \mathbf{N}), \tag{3}$$

where $\mathbf{A}_0$ is the unit vector along the incident ray, $\mathbf{A}_0^{'}$ is the unit vector along the refracted light, and $\mathbf{N}$ is the unit vector along the normal.

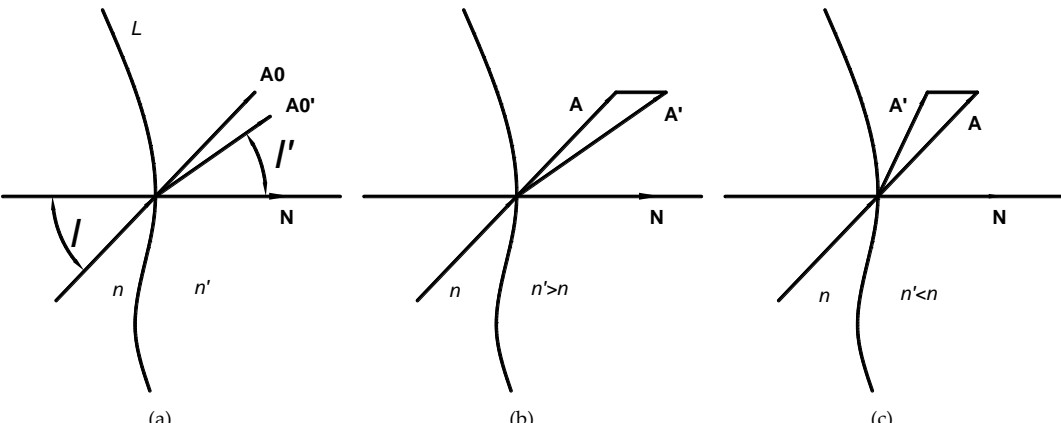

**Figure 1.** Schematic diagram of Snell's law in vector form: (**a**) Snell's law in vector form; (**b**) $\mathbf{A}'$-$\mathbf{A}$ and $\mathbf{N}$ are forward parallel; (**c**) $\mathbf{A}'$-$\mathbf{A}$ and $\mathbf{N}$ are antiparallel.

The refracted ray vector with length $n'$ and the incident ray vector with length $n$ are written as $\mathbf{A}^{'}$ and $\mathbf{A}$, respectively:

$$\mathbf{A}^{'} \times \mathbf{N} = \mathbf{A} \times \mathbf{N} \Rightarrow (\mathbf{A}^{'} - \mathbf{A}) \times \mathbf{N} = 0 \tag{4}$$

Formula (4) shows that vector $\mathbf{A}^{'}$-$\mathbf{A}$ and vector $\mathbf{N}$ are collinear vectors. Rewrite Formula (4) as:

$$\mathbf{A}^{'} - \mathbf{A} = P\mathbf{N}, \tag{5}$$

where $P$ is vector deflection constant. Product both sides of Formula (4) with $\mathbf{N}$ [6]:

$$P = \mathbf{N} \bullet \mathbf{A}' - \mathbf{N} \bullet \mathbf{A} = n' \cos I' - n \cos I, \tag{6}$$

when $n' > n$, the vectors $\mathbf{A}'$-$\mathbf{A}$ and $\mathbf{N}$ are parallel as shown in Figure 1b. When $n' < n$, the two vectors are antiparallel as shown in Figure 1c.

When the refractive index of the two media and the incident angle of the light are known, $n' \cos I'$ can be replaced with:

$$\begin{aligned} n' \cos I' &= \sqrt{(n')^2 - (n' \sin I')^2} \\ &= \sqrt{(n')^2 - n^2 + (n \cos I)^2} \end{aligned} \tag{7}$$

Therefore, the refraction law of vector form is [6]:

$$\mathbf{A}^{'} = \mathbf{A} + P\mathbf{N} = \mathbf{A} + (\sqrt{(n')^2 - n^2 + (n \cos I)^2} - n \cos I)\mathbf{N} \tag{8}$$

*2.2. Ray Tracing in the Gradient Refractive Index Medium*

The ray tracing method in the gradient refractive index medium proposed in this paper is based on the refractive index step. The tracing process is shown in Figure 2. In this figure, $n_0, n_1, n_2, n_3$ are the equal refractive index surfaces, and $n_3 = n_2 + t = n_1 + 2 \times t = n_0 + 3 \times t$ ($t$ is the refractive index step). The medium between $n_0$ and $n_1$ is regarded as an equal refractive index medium, and the value of refractive index is $\frac{n_0 + n_1}{2}$. Similarly, the refractive index of the medium between $n_1$ and $n_2$ is $\frac{n_1 + n_2}{2}$. The refractive index of the medium between $n_2$ and $n_3$ is $\frac{n_2 + n_3}{2}$. $P_0, P_1, P_2, P_3$ are the intersection points of ray $L$ and constant refractive-index surfaces $n_0, n_1, n_2, n_3$. When the ray $L$ enters the medium from point $P_0$ along direction $\mathbf{e}_{\mathbf{P}_0\mathbf{P}_1}$, its actual propagation path is $L_1'$. Since the medium

between $n_0$ and $n_1$ is regarded as an equal refractive index medium, the Line segment $L_1$ is approximately regarded as the equivalent propagation path of ray between $n_0$ and $n_1$ In the same way, the actual ray propagation path $L_2'$, $L_3'$, is approximately replaced with $L_2$, $L_3$, respectively.

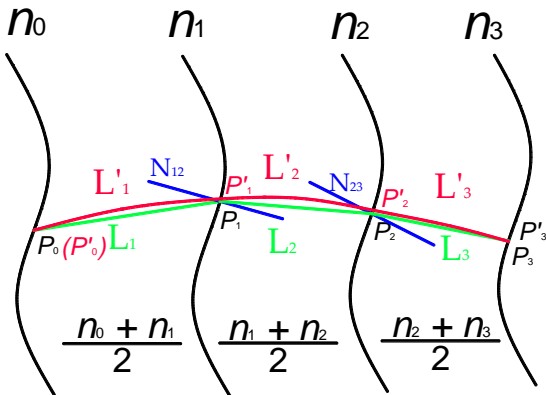

**Figure 2.** Schematic diagram of ray tracing in gradient index media.

The constant-index surface $n_i$ is regarded as refractive surface. According to Formula (8), the direction vector of the refracted ray can be expressed as:

$$\mathbf{e}_{\mathbf{p}_1\mathbf{p}_2} = \mathbf{e}_{\mathbf{p}_0\mathbf{p}_1} + P_{12}\mathbf{N}_{12}, \tag{9}$$

where $\mathbf{e}_{\mathbf{p}_1\mathbf{p}_2}$ and $\mathbf{e}_{\mathbf{p}_2\mathbf{p}_3}$ are the direction vectors of the incident ray and refractive ray, and the $\mathbf{N}_{12}$ is the normal of constant refractive-index surface. At this point, we can get the direction vector of the refracted ray. Since the refractive index step is used instead of the geometric step in this solution, the coordinate of the next refraction point $P_2$ cannot be determined directly according to $\mathbf{e}_{\mathbf{p}_1\mathbf{p}_2}$. Therefore, we propose to solve the coordinate of point $P_2$ by solving the combined formula of straight line $P_1P_2$ and the constant refractive-index surface $n_2$:

$$\begin{cases} F1 = f(x,y,z) \\ F2 = f'(x,y,z) \end{cases} \tag{10}$$

In the same way:

$$\mathbf{e}_{\mathbf{p}_2\mathbf{p}_3} = \mathbf{e}_{\mathbf{p}_1\mathbf{p}_2} + P_{23}\mathbf{N}_{23} \tag{11}$$

So far, the iterative formula, which is based on the refractive index step for ray tracing in gradient index media, can be obtained. According to the iterative formula, we can calculate the OPL (optical path length):

$$OPL = \sum n_i \times S_{P_iP_{i+1}}, \tag{12}$$

where $S_{P_iP_{i+1}}$ is the distance of $P_i$ and $P_{i+1}$.

## 3. Algorithm Verification

For most of the non-uniform refractive index medium, the analytical solution of the ray trajectory cannot be derived from the ray formula. However, the analytical solution can be acquired for some special refractive index distributions [1]. We compare the analytical solutions of two representative refractive index distributions with the numerical solutions obtained by the proposed algorithm. All the following ray tracing processes take the $+z$ axis as the optical axis direction, and in order to compare the tracing accuracy of RTRIS and RTSGS, the iterative steps of the two tracing methods are the same in the tracing process.

*3.1. Example 1*

The distribution function of the refractive index field is:

$$n = n_0 + \alpha z, \tag{13}$$

where $\alpha$ is the medium distribution constant. The linear function distribution of the axial gradient refractive index is the simplest form of axial gradient refractive index distribution. Under the condition of paraxial, the analytical solution of the ray formula is:

$$\begin{cases} x = x_0 + \frac{p_0}{\alpha} \ln\left[\frac{n_0 + \alpha z}{n_0 + \alpha z_0}\right] \\ y = y_0 + \frac{q_0}{\alpha} \ln\left[\frac{n_0 + \alpha z}{n_0 + \alpha z_0}\right] \end{cases} \tag{14}$$

where $p_0$, $q_0$ are the optical direction cosine of the ray. When $x = 0$, the constant $a = n_0$ is the refractive index of the incident surface. Since the refractive index of glass and optical fiber is generally around 1.5, so take $n_0 = 1.5$ and $\alpha = -0.0005$. When the incident point coordinate is (0, 0, 0) and incident ray direction vector is (0, 0.0001, 1), Formula (14) can be simplified as:

$$\begin{cases} x = 0 \\ y = \frac{0.0001}{-0.0005} \ln\left[\frac{1.5 - 0.0005 \times z}{1.5}\right] \end{cases} \tag{15}$$

When tracing length is 2000 mm, the optical path analytical solution is:

$$\begin{aligned} L1 &= \int_l n \, ds \\ &= \int_0^{2000} n \times \sqrt{1 + (\frac{0.0001}{-0.0005} \times \ln\left[\frac{1.5 - 0.0005 \times z}{1.5}\right])} \, dz \\ &= 2.000000025 \times 10^3 \text{mm} \end{aligned} \tag{16}$$

Use the algorithm proposed in this paper to trace the gradient index medium with a linear function of refractive index distribution from 0 to 2000 mm, we can obtain the optical path $L1 = 2.000000027 \times 10^3$. It can be found that the order of tracing accuracy can reach $10^{-6}$, and the accuracy can completely meet the demand of engineering accuracy.

In Figure 3, the RTRIS represents the error between the optical path obtained by the ray tracing method based on the refractive index step and the actual optical path from 0 to 2000 mm. In addition, the RTSGS represents the error between the optical path obtained by the ray tracing method based on the geometric step and the actual optical path. It is showing that when the tracing distance is in the range of 0 to 400 mm, the errors of the two tracing methods are almost the same, and the error value is small. When the tracing distance is longer than 400 mm, the error growth rate of RTRIS increases rapidly, and its error curve is similar to exponential curve. Finally, when the tracing distance is 2000 mm, the error value reaches $1.47 \times 10^{-5}$ mm. For RTSGS, when the tracing distance continues to increase, the error value increases slightly, and the growth rate is small. The error curve is flat. When tracing distance reaches 2000 mm, the error value is only $2.2 \times 10^{-6}$ mm, which is only 0.15 times as much as RTRIS. The primary source of error is the accumulation of computer errors and fitting errors of the constant refractive-index surfaces. It is showing that the ray tracing method based on the gradient refractive index step is more accurate and stable than that based on the space step for the gradient refractive index media distribution in accordance with Formula (11).

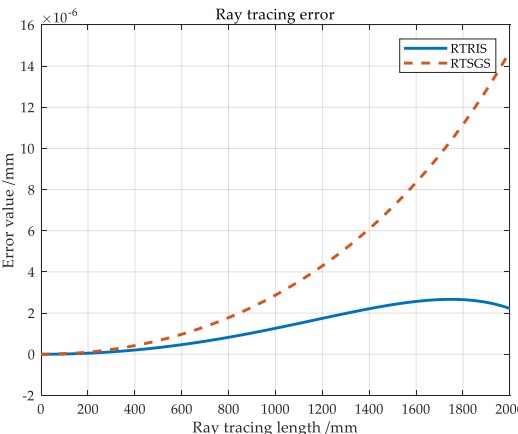

**Figure 3.** Ray tracing error example 1.

*3.2. Example 2*

The distribution function of refractive index field is:

$$n = \sqrt{n_0{}^2(1 - \alpha^2 z^2)} \tag{17}$$

The distribution function of the refractive index field shown in Formula (17) is one of the typical distribution forms that can be obtained by ion diffusion technology. When $n_0 = 1.5$ and $\alpha = -0.00049$, the refractive index distribution along the $z$ axis is shown in Figure 4. It can be seen that in the range of 0 to 1000 mm, the refractive index changes smoothly, and the gradient of the refractive index is small. As the tracing distance increases, the refractive index changes more and more rapidly, and the refractive index decreases rapidly. The gradient of the refractive index rises sharply. The refractive index distribution is extremely uneven.

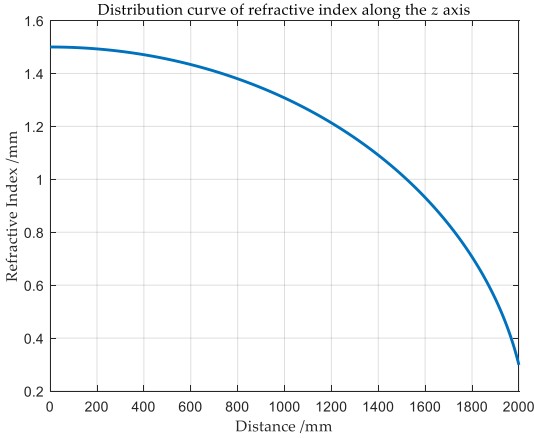

**Figure 4.** Distribution curve of the refractive index along the $z$ axis.

Under paraxial conditions, the analytical solution of the ray equation is:

$$\begin{cases} x = x_0 + \dfrac{p_0}{n_0 \times \alpha}[\arcsin(\alpha z) - \arcsin(\alpha z_0)] \\ y = y_0 + \dfrac{q_0}{n_0 \times \alpha}[\arcsin(\alpha z) - \arcsin(\alpha z_0)] \end{cases} \tag{18}$$

When the incident point coordinate is (0, 0, 0) and the incident ray direction vector is (0, 0.0001, 0), Formula (18) can be simplified as:

$$\begin{cases} x = 0 \\ y = \dfrac{0.0001}{1.5 \times (-0.0001)}[\arcsin(-0.0001z) - \arcsin(-0.0001z_0)] \end{cases} \tag{19}$$

When the tracing length is 2000 mm, the optical path analytical solution is:

$$
\begin{aligned}
L1 &= \int_l n \, ds \\
&= \int_0^{2000} n \times \sqrt{1 + \left(\frac{0.00049}{1.5 \times (-0.00049)} \times [\arcsin(-0.00049z) - \arcsin(-0.00049z_0)]\right)^2} \, dz \\
&= 2.39614158 \times 10^3 \, (\text{mm})
\end{aligned}
\tag{20}
$$

Using the algorithm proposed in this paper to trace the gradient refractive index medium from 0 to 2000 mm, it is possible to obtain the optical path length $L1 = 2.3961413 \times 10^3$.

The curve in Figure 5 shows the relationship between the geometric step length and the tracing distance of the two methods. The number of iterations for both of these two methods is 6821. For the traditional ray tracing method, the geometric step length is constant. For the method proposed in this paper, the geometric step length can be adjusted adaptively according to the refractive index distribution. In the first half of the tracing trajectory, the refractive index of the medium changes slower. Therefore, the geometric step size is larger. As the tracing trajectory gradually approaches 2000 mm, the refractive index of the medium changes intensify, and its geometric step size decreases rapidly to achieve higher accuracy.

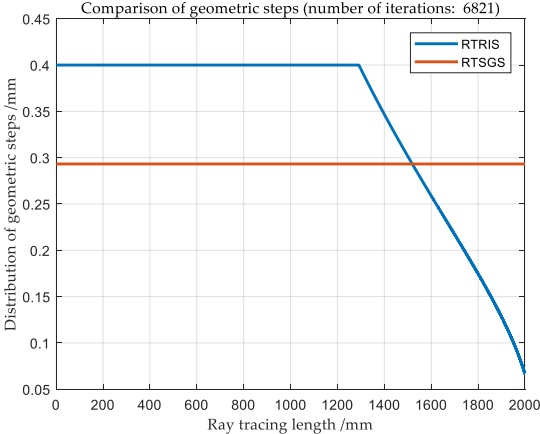

**Figure 5.** Comparison of geometric steps (number of iterations: 6821).

In Figure 6, the RTRIS represents the error between the optical path obtained by the ray tracing method based on the refractive index step and the actual optical path from 0 to 2000 mm. The RTSGS represents the error between the optical path obtained by the ray tracing method based on the geometric step and the actual optical path. Comparing the tracing errors of the two tracing methods in Figure 5, it can be found that the tracing accuracy of the ray tracing method based on the refractive index step is significantly better than that of the traditional ray tracing method based on the geometric step. When the tracing distance is 2000 mm, the cumulative error of the RTRIS is $2.5 \times 10^{-4}$ mm, and the cumulative error of the RTSGS is up to $3.13 \times 10^{-4}$ mm.

Combine Figures 3–5, in the initial stage of ray tracing, due to the slow change of the refractive index of the medium, the errors of the two methods are almost equal although the geometric step length of RTSGS is relatively large. As the tracing distance gradually increases, the advantages of the RTSGS gradually become apparent. Because the refractive index of the medium changes smoothly and the geometric step of RTSGS is larger, there are fewer iterations and smaller cumulative errors. When the tracing distance continues to increase, especially when it is close to 2000 mm, the refractive index of the medium decreases rapidly. The tracing step length for RTRIS cannot adaptively change, the error increases rapidly. RTSGS can adjust the geometric step length adaptively. Therefore, the geometric step is adjusted to be smaller in this section, which can better capture the refractive index change of the medium and improve the accuracy. It can be seen

that the essence of RTSGS is to reasonably allocate the geometric step length in the ray tracing process.

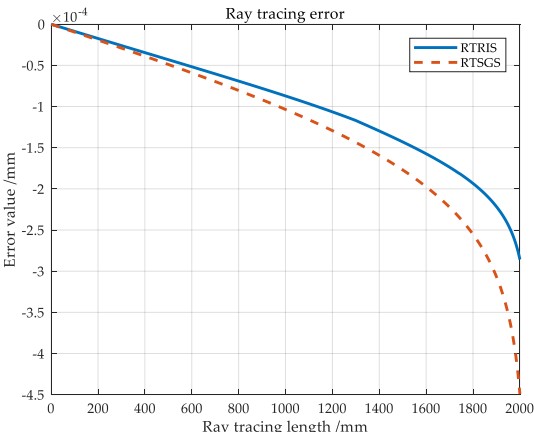

**Figure 6.** Ray tracing error example 2.

## 4. Conclusions

The ray tracing method based on the refractive index step proposed in this paper has extensive adaptability to the axial gradient refractive index, especially the inhomogeneous refractive index medium with a large refractive index gradient. It can efficiently solve the problem that the traditional ray tracing method based on geometric step cannot adapt to the inhomogeneous refractive index medium with large refractive index gradient or sudden change and improve ray tracing accuracy. At the same time, the proposed method can achieve high-efficiency ray tracing in a non-uniform refractive index medium with small refractive index change rate due to the refractive index step used in the recursion process. By using the proposed method, the efficiency of ray tracing can be achieved while the accuracy of ray tracing can be improved. Thus, the method realizes the adaptive process of the gradient index medium in the ray tracing process.

This method, however, has some problems which have to be solved. The most important one is that the constant refractive-index surfaces need to be fitted during the iterative process, so as to solve the iteration point through the fitting formula. The current method is to use high-order polynomials to fit the constant refractive-index surfaces, which might introduce a certain error. If the polynomial order is too high, it will cause Runge phenomenon. Therefore, different fitting models need to be considered in order to improve the ray tracing accuracy for different non-uniform refractive index media. Although there are some shortcomings in this method, we believe that this method will have a good application in ray tracing in gradient index media, especially in shock waves.

**Author Contributions:** Conceptualization, Q.Z.; and Y.T.; methodology, Q.Z.; software, Q.Z.; validation, Y.T., G.R. and T.T.; formal analysis, G.R.; investigation, Q.Z.; resources, Y.T.; data curation, T.T.; writing—original draft preparation, Q.Z.; writing—review and editing, T.T.; visualization, Q.Z.; supervision, T.T.; All authors have read and agreed to the published version of the manuscript.

**Funding:** This research was funded by Open Research Fund of State Key Laboratory of Pulsed Power Laser Technology (SKL2018KF05), Excellent Youth Foundation of Sichuan Scientific Committee (2019JDJQ0012), Youth Innovation Promotion Association, CAS (2018411), CAS "Light of West China" Program and Young Talents of Sichuan Thousand People Program.

**Institutional Review Board Statement:** Not applicable.

**Informed Consent Statement:** Not applicable.

**Data Availability Statement:** Not applicable.

**Conflicts of Interest:** The authors declare no conflict of interest.

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
