# Peer review of "Ray Tracing Method of Gradient Refractive Index Medium Based on Refractive Index Step"

_applsci, doi:10.3390/app11030912_

Round 1
Reviewer 1 Report
Strong aspects:
The paper proposes a method for ray tracing which is important for the calculation of the OPL when the light propagates in some conditions.
Weak aspects:
The paper is poorly written and the main ideas are difficult cu follow. It should be revised significantly for clarity.
More evaluation of the proposed method should be done.
The importance of ray tracing when it passes through shock layer of an aircraft is not very clear.
For how long the shock layer occurs ?
Conclusions section is missing.
Comments to the authors:
- row 32: noticed instead of notices
- The word 'And' should not start a sentence. Revise the paper to avoid the use of 'And' as the first word of a sentence.
- The word 'equation' should be used instead of 'formula' and the Eq. number should be put brackets
- The number for the refractive indexes (n) should be written as subscript.
- On row 100 between n2 and n3 the refractive index should be (n2+n3)/2 not (n1+n2)/2.
- In Equation 12 it is not clear what alfa, p0 and q0 represents.
- On row 113 double check if Eq 11 can be simplified by Eq. 14
- The measuring unit is missing for Eq. 15.
- It is not clear if n0 is n(0)
- The numbers on rows 165 and 166 should be aligned with the text.
- Avoid repetition on rows 194 and 195.
Author Response
Dear Reviewer:
We want to thank you for your constructive and insightful criticism and advice. Those comments are all valuable and very helpful for revising and improving our paper, as well as the important guiding significance to our researches. We modified the article based on your suggestion and the specific modification are in the attachment.
Thank you again for your valuable comments.
King regards,

Reviewer 2 Report
The manuscript discusses a method intended to speed up and to make more accurate ray tracing in media with gradient refractive index. In their method, the authors split media into slices of constant refractive index with a constant refractive index (n) step, rather than a common method of a constant distance or time steps. The authors derive their equations first and then compare solutions of two problems with the corresponding analytical solutions, using their method and a common ray tracing method. Apparently, their method gives smaller error.
The manuscript is written not clearly and in several places I cannot understand the meaning. I listed such places in the particular comments below. Further, it is not clearly stated if the same or similar media splitting method, i.e. splitting the media into constant n slices, has been already published in the literature.
In the solutions of the two example problems, the authors do not provide information on the values of the refractive index steps used. If these were constants, they can be written in the text. If they were adaptively selected, these can be shown in figures or in figure insets.
Another concern is that the authors state that their method is superior, showing the error comparison with the common method. However, I am sure that by using smaller step size in the common method it is possible to achieve the same or even smaller errors. Thus, it is necessary to compare errors obtained with the same calculation durations.
Finally, the most interesting problem to solve is may be ray tracing through a hydrodynamic shock, and that problem is totally absent in this manuscript.
Particular comments:
- The first names are often used instead of last names in the references.
- Equation (1) and the following discussion: the "s" quantity must be scalar, in contrast to the description. Also, it is a good practice to use bold fonts for vectors.
- There are unclear sentences, in particular lines 48, 50, 51.
- The word "recursive" is often used through the text, however, its assumed meaning seems to be different from the common one. In some places, a word "iterative" seems more appropriate, if I am catching the meaning correctly.
- The paragraph consisting of lines 69-72 is not clear.
- The description of derivation of equations (5), (6), and (7) is not clear and must be completely reworded.
- The equations (6) and (7) are identical to equations (6) and (7) of Reference [6]. This must be clearly stated in the manuscript.
- A misprint in the beginning of line 100: (n1+n2)/2 should read (n2+n3)/2.
- 2: generally speaking, the points P1, P2, and P3 used in the ray tracing are different from the points P'1, P'2, and P'3 which the real ray would go through. It might be better to show this difference in Fig.2.
- Error in equation (8): according to the standard notation, vectors P1P2 and P0P1 are the vectors with the starting points of P1 and P0 and the ending points of P2 and P1, respectively. Thus, they cannot be inserted into (7) because this equation assumes vectors with lengths equal to corresponding refractive indices. I believe, this error can be corrected by introducing direction vectors with appropriate names.
- Equation (11): the "cross" symbol was used earlier to denote the vector product (or at least I understood it like that); however, in this equation (as well as in the later equations) the same
"cross" symbol is used to denote simple product. That is very confusing.
- Again equation (11): the derivation is not clearly written. Further, the quantity L is not described well. Earlier, in lines 100 and 101, it was written that L is a ray. However, I cannot understand what kind of vector is it.
- Equation (13): the symbols p0 and q0 are not defined.
- Line 194 is repeated partly in Line 195.
Author Response
Dear Reviewer:
We want to thank you for your constructive and insightful criticism and advice. Those comments are all valuable and very helpful for revising and improving our paper, as well as the important guiding significance to our researches. We modified the article based on your suggestion and the specific modification are in the attachment.
Thank you again for your valuable comments.
Kind regards.

Round 2
Reviewer 1 Report
The paper was revised according to the suggestions. However the current Discussions section looks like a future work which can be presented as the last paragraph of the Conclusions section. Moreover, the Discussions section is not typically presented after the Conclusions.
Author Response
Dear Reviewer:
We want to thank you again for your constructive and insightful criticism and advice. We have put the discussion section into the conclusion as the last paragraph. The section of conclusion is as follows:
The ray tracing method based on refractive index step proposed in this paper has extensive adaptability to the axial gradient refractive index, especially inhomogeneous refractive index medium with large refractive index gradient. It can efficiently solve the problem that the traditional ray tracing method based on geometric step cannot adapt to the inhomogeneous refractive index medium with large refractive index gradient or sudden change and improve ray tracing accuracy. At the same time, the proposed method can achieve high-efficiency ray tracing in a non-uniform refractive index medium with small refractive index change rate due to the refractive index step used in the recursion process. By using the proposed method, the efficiency of ray tracing can be achieved while the accuracy of ray tracing can be improved. Thus, the method realizes the adaptive process of the gradient index medium in the ray tracing process.
This method, however, has some problems which have to be solved. The most important one is that the constant refractive-index surfaces need to be fitted during the iterative process, so as to solve the iteration point through the fitting formula. The current method is to use high-order polynomials to fit the constant refractive-index surfaces, which might introduce a certain error. If the polynomial order is too high, it will cause Runge phenomenon. Therefore, different fitting models need to be considered in order to improve the ray tracing accuracy for different non-uniform refractive index media. Although there are some shortcomings in this method, we believe that this method will have a good application in ray tracing in gradient index media, especially in shock waves.
Thank you again for your valuable comments.
Reviewer 2 Report
The authors improved the manuscript, answering most of my comments, although not all. There are a few remaining issues:
In my previous review I wrote " 7. The equations (6) and (7) are identical to equations (6) and (7) of Reference [6]. This must be clearly stated in the manuscript." The authors wrote in the cover letter, that they added [6] to this section, however, I cannot see this. Instead, the authors refer to [18] in this section, which is a later paper, (2015) compared to (1982) of [6].
Figure 5 caption: information on total number of iterations for RTRIS and RTSGS methods should be provided.
Line 183: Figure 5 -> Figure 6.
Author Response
Dear Reviewer:
We want to thank you again for your constructive and insightful criticism and advice. We have modified the article again based on your suggestion and the specific modification are as follows:
- In my previous review I wrote " 7. The equations (6) and (7) are identical to equations (6) and (7) of Reference [6]. This must be clearly stated in the manuscript." The authors wrote in the cover letter, that they added [6] to this section, however, I cannot see this. Instead, the authors refer to [18] in this section, which is a later paper, (2015) compared to (1982) of [6].
On row 78, we rewrite the sentence as “According to the reference [6], we can calculate Snell's law in vector form.” to indicates that we quote the formula in reference 6.
We also add [6] to row 89 and 94. The modify details is list as follows:
On row 89, we rewrite the sentence as “Product both sides of formula (4) with [6]:”.
On row 94, we rewrite the sentence as “Therefore, the refraction law of vector form is [6]:”.
- Figure 5 caption: information on total number of iterations for RTRIS and RTSGS methods should be provided.
The number of iterations for both of RTRIS and RTSGS methods is 6821. We add this information to the caption of Figure5 and row 178 of the article.
- Line 183: Figure 5 -> Figure 6.
We have modified “Figure 5” to “Figure 6” on row 183
Thank you again for your valuable comments.
